# Establishment of two pathomic-based machine learning models to predict CLCA1 expression in colon adenocarcinoma

**Caiyun Yao, Maotong Hu, Lingxia Zhou, Hui Chen, Yang Cao** *

Pathology Department, Yiwu Central Hospital, Jinhua, Zhejiang, China

* 1814218039@qq.com

## Abstract

Chloride channel accessory 1 (CLCA1) is considered a potential prognostic biomarker for colon adenocarcinoma (COAD). The objective of this research was to develop two pathomics models to predict CLCA1 expression from hematoxylin-eosin (H&E) stained pathological images and to investigate the biological mechanisms linked to pathomics features by associating the pathomics model with transcriptomic data. The prognostic value of CLCA1 in COAD was assessed based on gene transcriptome expression data. The two pathomics models were constructed to predict CLCA1 expression in COAD based on pathological image features using the random forest (RF) and XGBoost machine learning algorithms. The RF pathomics model demonstrated superior predictive performance, achieving area under the curve (AUC) values of 0.846 and 0.776 in the training and validation cohorts, respectively, and was selected for further analysis. The ability of the pathomics model to predict overall survival (OS) in COAD was determined using univariate and multivariate Cox regression analyses. The possible biological mechanisms behind the pathomics model were explored by conducting gene set variation analysis (GSVA), immune infiltration assessment, and somatic mutation analysis. CLCA1 expression was downregulated in COAD patients and was associated with a poor prognosis ($P = 0.008$). Participants were categorized into high- and low-risk score groups based on the critical value of the risk score. High-risk scores were protective for OS in COAD in both univariate and multivariate Cox regression analyses. Meanwhile, GSVA enrichment analysis unveiled notable enrichment of pathways such as the epithelial-mesenchymal transition and vascular endothelial growth factor (VEGF) signaling in the low-risk score group. Two pathomics-based machine learning models were developed to predict CLCA1 expression from H&E stained images of COAD. A theoretical basis for interpreting the disease model was developed by comprehensively analyzing the pathomics-based models and transcriptomic data, facilitating further hypothesis-driven experimental research.

**Data availability statement:** All relevant data are within the manuscript and its Supporting information files.

**Funding:** The author(s) received no specific funding for this work.

**Competing interests:** The authors have declared that no competing interests exist.

**Abbreviations:** CLCA1, Chloride channel accessory 1; COAD, Colon adenocarcinoma; H&E, Hematoxylin-eosin; RF, Random forest; AUC, Area under the curve; OS, Overall survival; GSVA, Gene set variation analysis; VEGF, Vascular endothelial growth factor; CLCAs, Calcium-activated chloride channels; EMT, Epithelial-mesenchymal transition; TCGA, The cancer genome atlas; RNA-seq, RNA sequencing; K-M, Kaplan-Meier; RFE, Recursive feature elimination; ROC, Receiver operating characteristic; DCA, Decision curve analysis; TPR, True positive rate; PR, Precision-recall; HL, Hosmer-Lemeshow; KEGG, Kyoto encyclopedia of genes and genomes; HR, Hazard ratio; CI, Confidence interval; KRAS, Kirsten rat sarcoma viral oncogene homolog; CD27, Tumor necrosis factor receptor superfamily member 7; CD40LG, CD40 ligand gene; ICOS, Inducible co-stimulator; TMIGD2, Transmembrane and immunoglobulin domain containing 2; NK, Natural killer; APC, Adenomatous polyposis coli; TP53, Tumor suppressor protein p53; TTN, Titin gene

## Introduction

Colon adenocarcinoma (COAD) ranks as the third most frequent malignant gastrointestinal cancer and the second leading cause of cancer mortality worldwide, with an estimated 2.2 million new cases and 1.1 million deaths by 2030 [1]. Its aggressive progression, high fatality rate, and unfavorable prognosis pose a significant public health threat [2]. Currently, therapeutic approaches for COAD primarily involve surgical intervention, chemotherapy or radiotherapy, targeted therapy, and immunotherapy. Despite a twofold increase in overall survival (OS) among patients with nonmetastatic COAD, the prognosis for advanced cases remains sub-optimal [3]. Notably, the efficacy of COAD treatment is limited by systemic toxicity and adverse effects associated with chemotherapy, as well as inherent or acquired resistance to targeted therapies. Ultimately, most patients who undergo surgery or chemotherapy experience relapse or metastasis [4,5]. Currently, effective biomarkers to predict the advancement of COAD and patient outcomes are still missing in clinical practice.

COAD pathogenesis is still being investigated, and it is widely acknowledged that epigenetic, dietary, environmental, and metabolic factors play a role in its development [6]. Ion channels are essential in the gastrointestinal tract for regulating various cellular and tissue processes. Moreover, they are frequently dysregulated in malignant tumors such as colon cancer and pancreatic cancers [7]. Calcium-activated chloride channels (CLCAs) represent a class of autocrine proteins that activate calcium-dependent chloride currents in mammalian cells [8]. The CLCA proteins encoded by the human genome include chloride channel accessory 1 (CLCA1), CLCA2 and CLCA4 [9]. Notably, CLCA1 is essential for regulating chloride flow in epithelial cells. It is implicated in the pathogenesis of respiratory and gastrointestinal diseases associated with excessive mucus production, thereby impacting various cellular processes, including epithelial fluid secretion, mucus synthesis, signal transduction, cell adhesion, cell cycle regulation, cell apoptosis, tumorigenesis, and metastasis. There is growing evidence that it can be used as a cancer therapeutic target [10]. CLCA1 is widely recognized as a gene that inhibits tumors in several cancers despite limited studies and its nascent development phase [11]. Earlier studies on CLCA1 in tumors have primarily focused on colorectal [12], pancreatic [13], and ovarian cancers [14]. Knocking out CLCA1 in human colorectal adenocarcinoma caco-2 cells exerted inhibitory effects on cell differentiation and promoted proliferation concurrently [15]. *In vitro* studies suggest that CLCA1 can act as a tumor suppressor in COAD by hindering the Wnt/beta-catenin signaling pathway and preventing epithelial-mesenchymal transition (EMT) [16]. CLCA1 is involved in the growth, movement, and spread of metastatic COAD cells by regulating the calcium-activated transmembrane protein 16A [17]. CLCA1 may be a biomarker for the diagnosis, prognosis, and treatment of patients with COAD based on existing evidence. Currently, CLCA1 expression level can only be determined by the quantification of peripheral blood cytokines, tissue mRNA, and tissue protein (such as flow cytometry, immunohistochemistry, western blotting, among others). However, these methods are constrained by operator variability, availability of antibodies and cost. Therefore, finding efficient and convenient detection approaches is crucial.

Hematoxylin-eosin (H&E) stained slides are essential for clinical diagnosis as they are a reliable, cost-effective method. Currently, pathologists are unable to predict CLCA1 expression levels or other biomarkers using H&E stained images. The emergence of artificial intelligence technology has a profound impact on conventional pathological diagnosis and prognosis prediction [18]. Pathomics involves converting tissue pathology images into quantitative features using artificial intelligence, with applications in diagnosis, molecular data expression, and prognosis prediction [19]. This study utilized pathological images stained with H&E to establish two pathomics models for predicting CLCA1 expression in COAD. Moreover, pathology models and transcriptomic data were correlated to determine the biological mechanisms underlying pathomics characteristics.

## Materials and methods

### Data sources

This study used clinical data sourced from The Cancer Genome Atlas (TCGA) database (https://tcga-data.nci.nih.gov/), encompassing 459 COAD cases and 453 H&E stained pathology images. The criteria for inclusion were: (1) Patients with a primary condition, first diagnosis, and initial treatment; (2) COAD diagnosis verified by pathology; (3) RNA sequencing (RNA-seq) data availability. Criteria for exclusion were: (1) lack of survival data or survival duration under 30 days; (2) missing clinical information regarding tumor stage and grade; (3) substandard quality of pathological images (S1 and S2 Tables). Data on transcriptome expression and prognostic details for 33 tumor types were obtained from UCSC Xena (https://xenabrowser.net/datapages/). Additionally, RNA-seq data in fragments per kilobase million format were retrieved from the TCGA database.

### Expression differences and prognosis analysis of CLCA1

The differences in CLCA1 expression levels between healthy and cancerous tissues in 33 different tumor types were determined by the Wilcoxon tests. The minimum $P$-value method was used to calculate the critical value of CLCA1 expression for each patient using the R package "Survminer" (version 3.6.3). The Benjamini-Hochberg adjustment was applied to account for multiple testing. All patients were divided into two groups base on the critical value: those with high CLCA1 expression and those with low CLCA1 expression. Subsequently, Kaplan-Meier (K-M) survival curves were generated to depict differences in survival rates across various groups, with the median survival time representing the survival time corresponding to a 50% survival rate. The factors affecting OS in colorectal adenocarcinoma patients were determined by using the Cox proportional hazards model in a univariate correlation analysis. In addition, the independent and combined effects of several influencing factors on OS were investigated using multivariate analyses.

### Pathological image preprocessing

Pathological images were processed based on a combination of automatic and manual evaluation. For the first task, the OTSU method (https://opencv.org/) was used to divide the image into two sections: the irrelevant background and the tissue region essential for the study. The 40× images were segmented into multiple sub-images of 1000×1000 pixels, whereas 20× images were divided into multiple sub-images with 500×500 pixels. These sub-images were subsequently up-sampled to a resolution of 1000×1000 pixels. The pathological images underwent segmentation for manual evaluation and were subsequently reviewed by pathologists. Sub-images exhibiting poor image quality (such as contamination, blurring, or containing more than 50% blank area) were excluded from the analysis. From each pathological image, ten sub-images were selected randomly for further analysis.

### Extraction and screening of pathological images for feature analysis

Several CellProfiler software (https://cellprofiler.org/) components were used to develop an image analysis workflow for segmenting cell nuclei and extracting features. Initially, the "UnmixColors" module was utilized to process images and

adjust their resolution to 1000 × 1000 pixels, thereby enhancing accuracy in subsequent analysis. The adjusted images were subsequently subjected to automated segmentation using the "IdentifyPrimaryObjects" and "IdentifySecondaryObjects" modules, enabling precise differentiation of the nucleus and cytoplasm based on their respective positions and structures. In addition, various modules, such as "Object Intensity Distribution", were utilized to extract and quantify key features such as the shape, size, texture, and pixel intensity distribution of objects in images. This approach provided more comprehensive and detailed results for image analysis. After eliminating unnecessary image features, a total of 353 quantitative image features were ultimately selected for further analysis. To normalize pathomics features and remove discrepancies between their values, the StandardScaler function from the scikit-learn library was used. The subsequent step involved normalizing the validation set data using the mean and standard deviation calculated from the training set. Recursive feature elimination (RFE) was used to eliminate redundancy and avoid overfitting to select the optimal set of pathological features,.

## Construction and evaluation of the pathomics model

The two sets of pathomics models for predicting CLCA1 expression were constructed based on the selected pathological features using the random forest (RF) and XGBoost algorithms. The former, extensively evaluated through the combination of multiple decision trees, is associated with increased generalization error concentrations with an increase in the number of trees [20,21]. While, the XGBoost algorithm is a machine learning technique with the notable feature of assembling weak predictive models to build accurate models [22]. The predictive accuracy and clinical utility of models were evaluated using receiver operating characteristic (ROC) curves, calibration plots, and decision curve analysis (DCA) with the help of pROC, rms, rmda, measures, and ResourceSelection software tools. In the ROC curve, the horizontal axis denotes the false positive rate, whereas the vertical axis indicates the true positive rate (TPR). A larger ROC-area under the curve (AUC) value reflects a greater AUC, while a more pronounced curve bulging towards the upper left corner indicates superior model performance. In the precision-recall (PR) curve, the horizontal axis denotes recall, which is the TPR, while the vertical axis indicates precision. PR-AUC is an average of accuracies determined for each recall threshold. A more convex PR curve in the upper right corner suggests better performance. The calibration level of the predictive models was assessed using calibration curves and Hosmer-Lemeshow (HL) goodness-of-fit tests. Finally, DCA was used to display the practical value of the pathomics prediction model in a clinical setting.

## Analysis of the biological significance of pathomics models

The pathomics score (risk score) for each patient derived from the pathomics model was computed, and the threshold for this score was established utilizing the "Survminer" package. The risk score was then categorized into low/high binary classification variables for further analysis. The link between risk score and OS in patients with COAD was investigated using survival analysis techniques such as K-M survival curves, univariate analysis, and multivariate analysis. Additionally, gene set variation analysis (GSVA) was conducted utilizing the "GSVA" package in R, referencing the kyoto encyclopedia of genes and genomes (KEGG) (c2.cp.kegg.v7.5.1.symbols.gmt) and hallmark (hall.v7.5.1.symbols.gmt) data collections. The Wilcoxon test was employed to analyze the differential expression of immune genes between the high- and low-risk score groups. Subsequently, the gene expression matrix of the COAD sample was uploaded to the CIBERSORTx platform (http://cibersortx.stanford.edu/), which used the CIBERSORT algorithm to estimate the abundance of immune cell infiltration, and was based on the methodology employed by some researchers for assessing tumor-infiltrating lymphocyte subsets [23,24]. Immune cell infiltration levels were compared between groups with high- and low-risk scores using the "limma" R package (version 3.6.3). Ultimately, the frequency of somatic mutations in the high- and low-risk score groups was evaluated with the R package "maftools" (version 3.6.3).

## Statistical analyses

Categorical data (percentages) were analyzed using the chi-square test, whereas continuous data (Q1, Q3) were evaluated with the Wilcoxon rank-sum test. Survival rates were compared using the Log-rank test, and the HL test was used to assess the pathomics models using the R package (version 3.6.3). Statistical analyses and visualizations were conducted using the R package (version 3.6.3), and two-sided *P*-values less than 0.05 were considered statistically significant.

## Results

### CLCA1 is lowly expressed in COAD and is associated with poor prognosis

To begin, CLCA1 expression levels were analyzed between healthy and tumor tissues across 33 human tumor types (Fig 1A). According to the differential analysis results, CLCA1 expression was lower in various cancer tissues, including COAD, (*P* < 0.001). The transcriptome expression data from 332 COAD samples were included for analysis based on the inclusion criteria. Additionally, the samples were divided into high expression (n = 169) and low expression (n = 163) groups based on the CLCA1 expression threshold of 3.538. The distribution of different clinical variables across these groups was analyzed (S3 Table). High expression of CLCA1 was significantly associated with improved OS according to the K-M survival curves (Fig 1B). The univariate Cox analysis revealed that elevated CLCA1 levels were associated with a positive

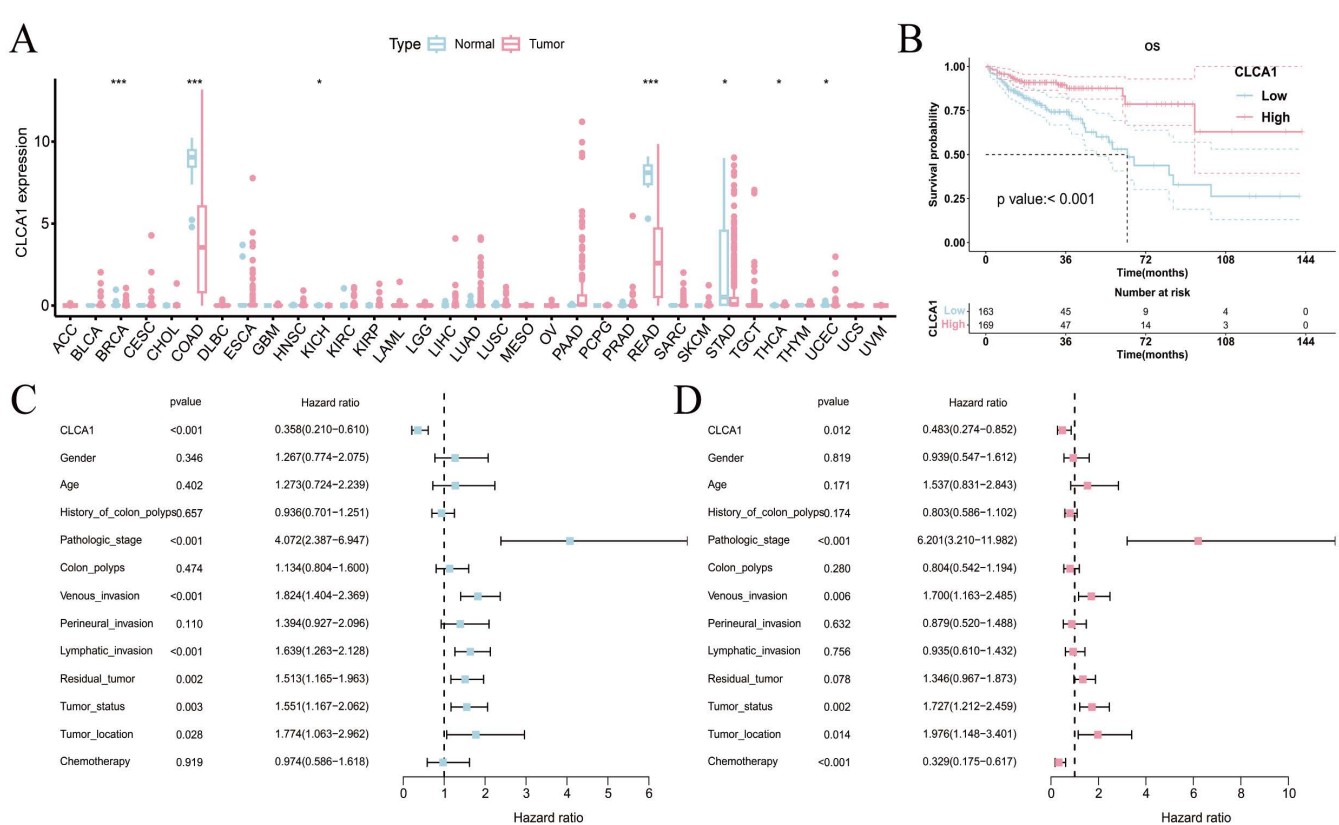

**Fig 1. Analysis of the prognostic value of CLCA1.** (A) differential expression analysis of CLCA1 in pan-cancer; (B) K-M survival curves plot; (C) unifactorial Cox analysis; (D) multifactorial Cox analysis. *P < 0.05; **P < 0.01; ***P < 0.001.

prognosis for OS (hazard ratio [HR] = 0.358, 95% confidence interval [CI]: 0.210–0.610, $P<0.001$) (Fig 1C). After adjusting for multiple factors, elevated CLCA1 levels continued to depict a significant correlation with better OS (HR = 0.483, 95% CI: 0.274–0.852, $P=0.012$) (Fig 1D).

## Development and evaluation of pathomics models for predicting the expression level of CLCA1 in COAD

According to the inclusion and exclusion criteria, 297 samples with pathological images were analyzed. The patients were randomly divided into training (n = 90) and validation (n = 207) groups with a 7:3 ratio. The analysis of inter-group differences between the training and validation sets revealed that the baseline characteristics of the patients in both sets were similar, suggesting that the groups are comparable (S4 Table). Pathological image processing and feature extraction yielded a total of 353 quantitative image features. Then, five optimal feature subsets were selected through the RFE algorithm. The RFE method demonstrated a positive correlation was found between Mean_IdentifyPrimaryObjects_Texture_AngularSecondMoment_Hematoxylin_3_00_256 (r = 0.028) and CLCA1 gene expression, while a notably negative correlation was detected between Mean_IdentifySecondaryObjects_Texture_Entropy_Hematoxylin_3 _01_25 (r = −0.077) and CLCA1 gene expression (Fig 2A). The significance of feature selection in the algorithm for the two pathomics models is displayed in (Fig 2B, C). The RF model demonstrated superior predictive performance among the two pathomics models. According to the ROC curve, the AUC for the training set was 0.846 (95% CI: 0.792–0.9), whereas the validation set had an AUC of 0.776 (95% CI: 0.674–0.878). The calibration curve and the HL goodness-of-fit test demonstrated that the predicted probability of CLCA1 expression in the training and validation datasets aligned with the observed outcomes ($P=0.417$ and 0.468, respectively). Furthermore, DCA analysis demonstrated the high clinical applicability of the model (Fig 3A–H). Regarding performance indicators, the training set showed an accuracy of 0.812, a sensitivity of 0.88, a specificity of 0.722, and a Brier score of 0.191. Correspondingly, in the validation set, these indicators

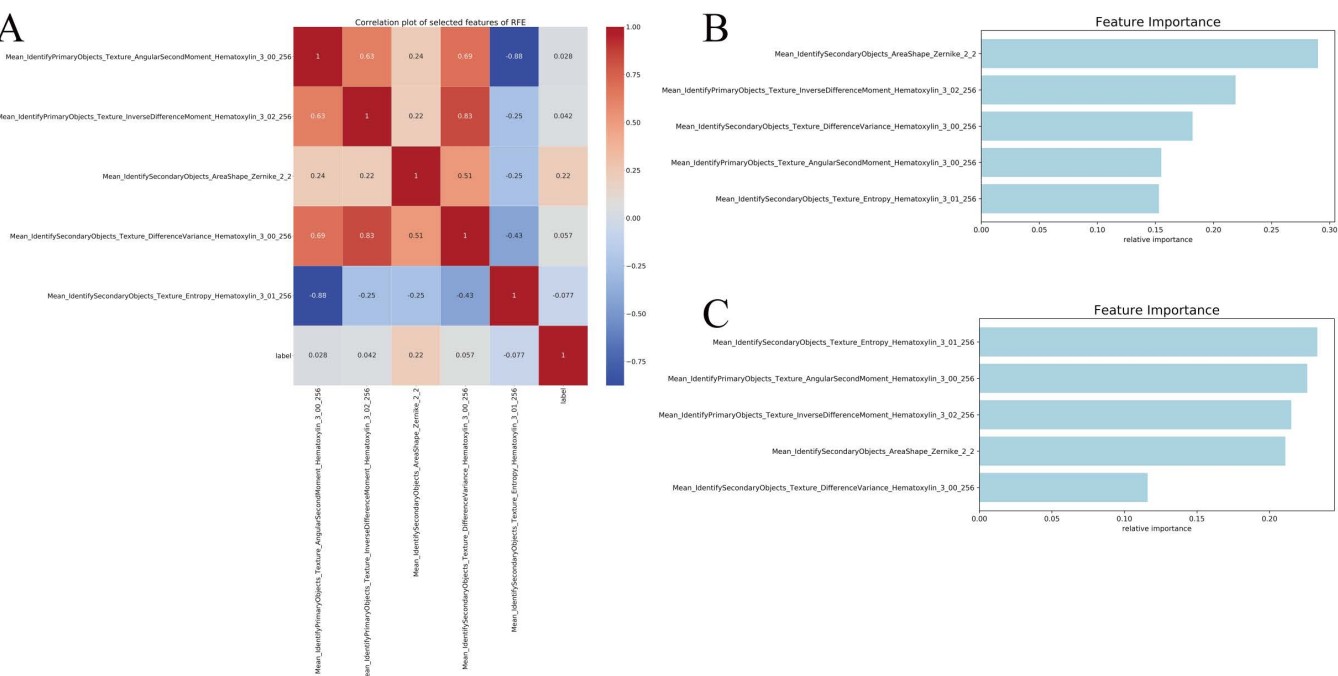

**Fig 2. Feature Screening.** (A) heatmap visualization of the correlation coefficient matrix for five pathohistological features; (B) importance of features in the RF algorithm; (C) importance of features in the XGBoost algorithm.

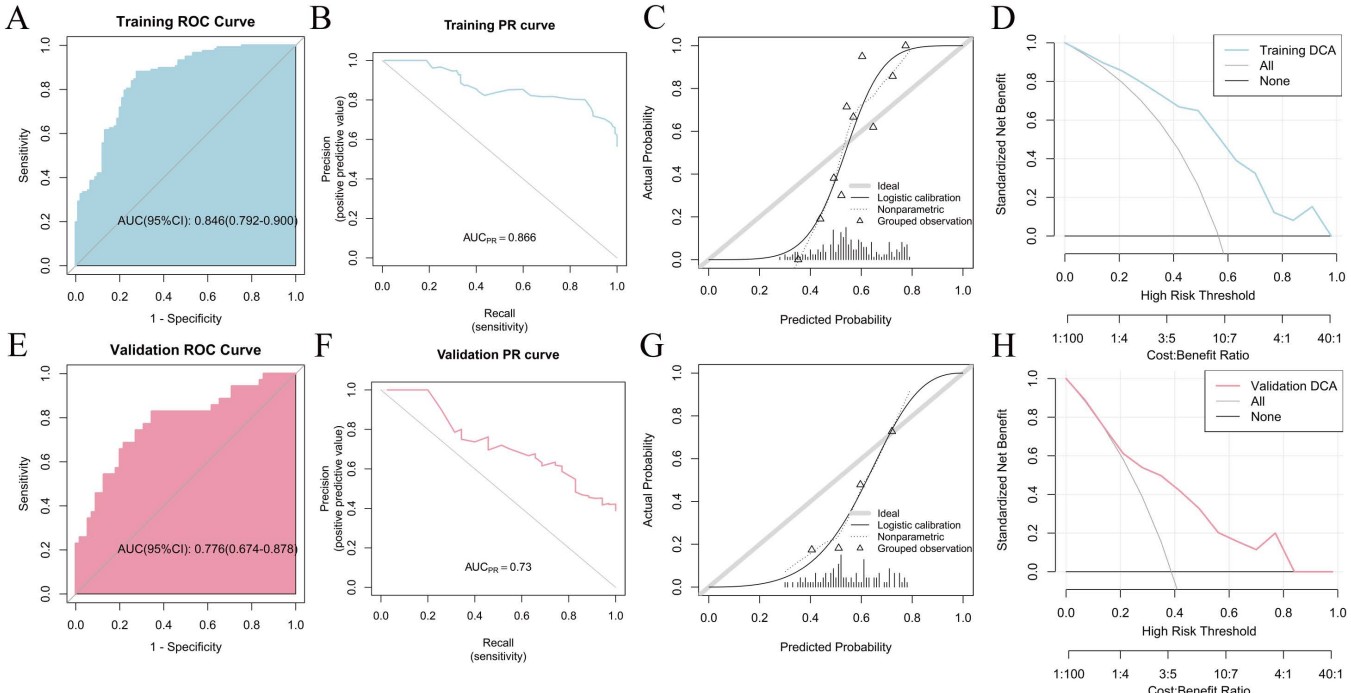

**Fig 3. Evaluation of the RF model.** (A) Training set ROC curve; (B) Training set PR curve; (C) Training set calibration curve; (D) Training set DCA curve; (E) Validation set ROC curve; (F) Validation set PR curve; (G) Validation set calibration curve; (H) Validation set DCA curve.

were 0.722, 0.829, 0.655, and 0.225, respectively (S5 Table). The XGBoost model achieved an AUC of 0.819 (95% CI: 0.763–0.876) on the training set and 0.789 (95% CI: 0.689–0.888) on the validation set. Similarly, both the training and validation datasets were in agreement with the estimated likelihood of CLCA1 expression, as shown by the calibration plots and HL fit tests ($P > 0.05$). Additionally, the DCA curve highlighted the significant clinical utility of the model (Fig 4A–H). The accuracy, sensitivity, specificity, and Brier scores were 0.739, 0.641, 0.867, and 0.192 in the training set, whereas they were 0.767, 0.771, 0.764, and 0.222, respectively, in the validation set (S6 Table).

## Prognostic evaluation of the pathomics models

The study indicated that the risk score was notably elevated in the group with high CLCA1 expression compared to the group with low expression across both pathomics models ($P < 0.05$) (Fig 5A–D). The RF model was selected for subsequent analysis of the predicted results due to its superior performance in evaluation indicators such as AUC and accuracy when compared to the XGBoost model. Besides, 297 patients were divided into high- (n = 176) and low-risk (n = 121) groups based on a risk score threshold of 0.527 that was determined using the "Survminer" package (S7 Table). According to K-M survival curves, patients with high-risk scores had a higher OS than those with low-risk scores($P = 0.008$), suggesting that elevated risk scores are linked to a more favorable prognosis in patients with COAD (Fig 6A). In a similar vein, a high-risk score was associated with better OS in COAD (HR = 0.507, 95% CI: 0.304–0.846, $P = 0.009$) and (HR = 0.361, 95% CI: 0.196–0.665, $P = 0.001$) as depicted by the univariate and multivariate Cox regression analyses (Fig 6B, C).

## The potential biological mechanisms of the pathomics model

The potential biological mechanisms of the pathomics model were further elucidated through GSVA functional enrichment analysis. Analysis of KEGG gene sets analysis indicated a notable enrichment of the NOD-like receptor signaling pathway,

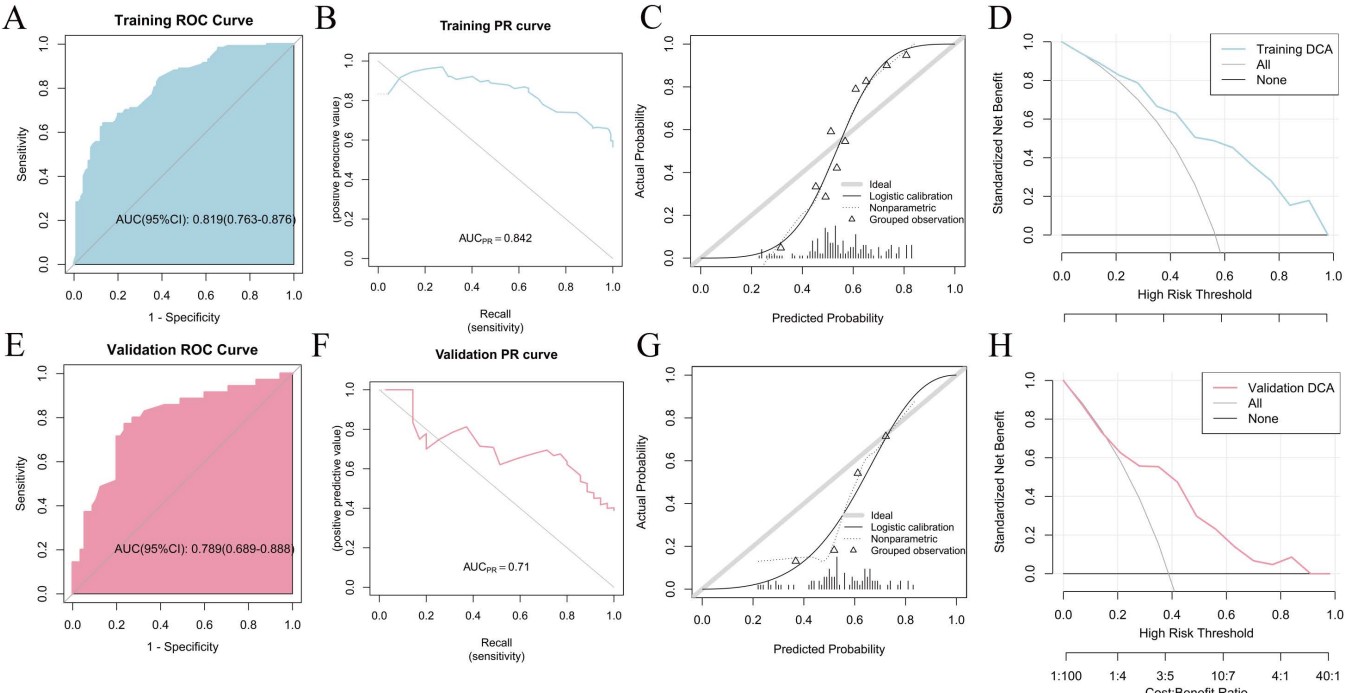

**Fig 4. Evaluation of the XGBoost model.** (A) Training set ROC curve; (B) Training set PR curve; (C) Training set calibration curve; (D) Training set DCA curve; (E) Validation set ROC curve; (F) Validation set PR curve; (G) Validation set calibration curve; (H) Validation set DCA curve.

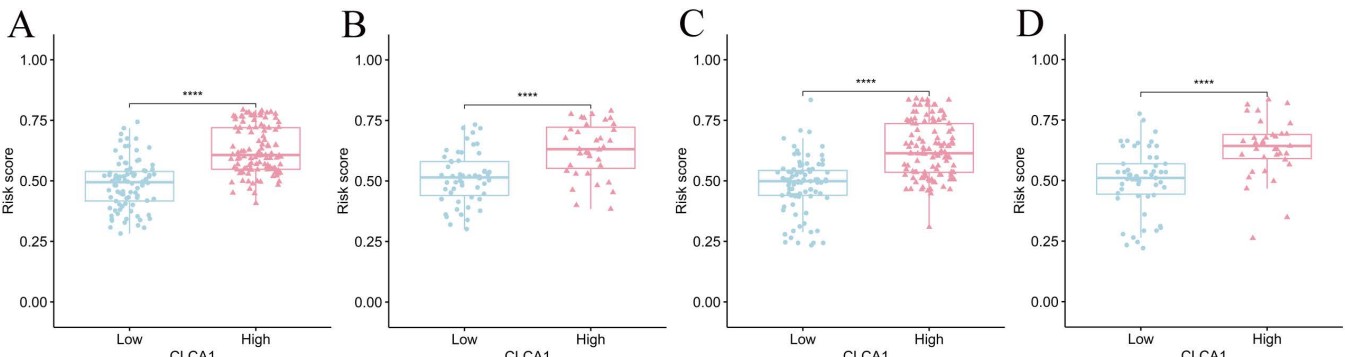

**Fig 5. Distribution of risk score output from different pathomics models between high and low CLCA1 gene expression groups.** (A) distribution of risk score of RF model (Training set) between high and low CLCA1 gene groups; (B) distribution of risk score of RF model (Validation set) between high and low CLCA1 gene groups; (C) distribution of risk score of XGBoost model (Training set) between high and low CLCA1 gene groups; (D) distribution of risk score of XGBoost model (Validation set) between high and low CLCA1 gene groups. ****$P<0.0001$.

the Janus kinases-signal transducer and activator of transcription pathway, and the vascular endothelial growth factor (VEGF) signaling pathway in the subgroup with a low-risk score (Fig 7A).

Similarly, the outcomes from the hallmark gene set analysis revealed a notable enrichment of the EMT, interleukin-6/janus kinases/signal transducer and activator of transcription 3, kirsten rat sarcoma viral oncogene homolog (KRAS), tumor necrosis factor-α, and hypoxia signaling pathways in the low-risk score group (Fig 7B).

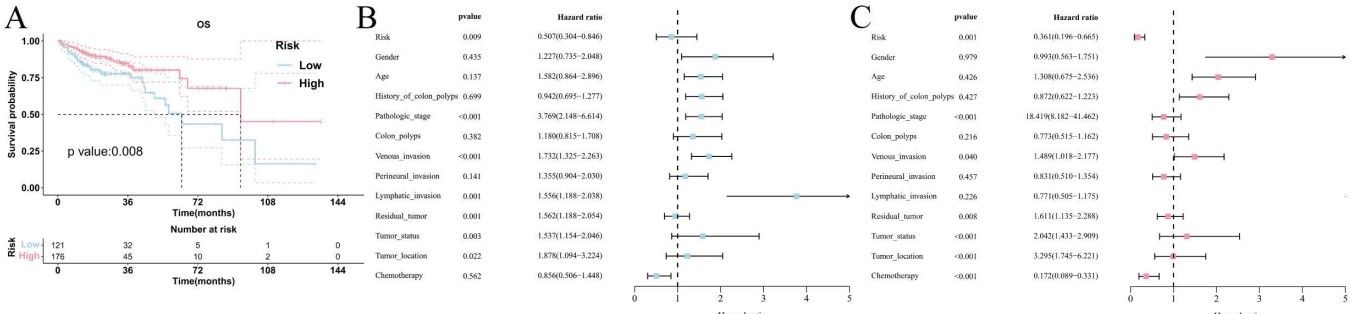

**Fig 6. Analysis of the prognostic value of the risk score.** (A) K-M survival curves for the analysis of high- and low-risk score versus OS; (B) unifactorial Cox analysis; (C) multifactorial Cox analysis.

Additionally, the immune gene expression analysis revealed that the levels of tumor necrosis factor receptor superfamily member 7 (CD27), CD40 ligand gene (CD40LG), inducible co-stimulator (ICOS), and transmembrane and immunoglobulin domain containing 2 (TMIGD2) were significantly elevated in individuals with high-risk scores (Fig 7C).

Notably, the presence of plasma cells, eosinophils, M2 macrophages, and inactive mast cells showed a positive correlation with elevated risk scores. Conversely, the infiltration levels of M0 macrophages and resting natural killer (NK) cells revealed a positive correlation with the low-risk scores (Fig 8A). According to mutation analysis results, missense mutations were most common, followed by multiple mutations and nonsense mutations. Both high- and low-risk score groups exhibited mutation frequencies exceeding 40% for adenomatous polyposis coli (APC), tumor suppressor protein p53 (TP53), titin gene (TTN), and KRAS, although TP53 and KRAS mutations were less frequent in the high-risk group. (Fig 8B, C).

## Discussion

In this study, two pathomics models were established to predict CLCA1 expression in COAD based on H&E stained pathological images. Associating the pathomics model with transcriptomic profiles enabled the exploration of the potential biological mechanisms behind pathomics features, providing valuable insights for the clinical diagnosis and treatment of COAD.

Reduced CLCA1 expression in tumors has been significantly linked to poor outcomes in several human cancers, such as colorectal, ovarian, pancreatic, and liver cancers [14,25–27]. Pan-cancer studies offer a deeper understanding of the fundamental mechanisms underlying the initiation and progression of cancer [28]. The differential analysis revealed the downregulation of CLCA1 expression in tumor tissues from multiple tumors, including COAD. The survival analysis revealed a positive correlation between low CLCA1 expression levels and shorter OS. Consequently, CLCA1 is likely to serve as a prognostic indicator for COAD. However, Chen et al. reported that patients with high CLCA1 expression undergoing radiotherapy and chemotherapy had a worse outcome [29]. The discrepancy can be attributed to several factors, including patient characteristics, sample size, and immunohistochemistry staining grades.

Recently, there has been significant progress in identifying molecular markers, predicting outcomes, and developing molecular biology and artificial intelligence technologies. Particularly, the successful application of pathomics in tumors, including automatic diagnosis, risk assessment, and survival prediction, has yielded promising results in patient prognosis [30,31]. Notably, it is commonly used to evaluate microsatellite instability, predict lung metastasis prognosis, and predict pathological staging in colon cancer [32–34]. These studies collectively indicate that digital pathomics signatures can potentially reflect underlying molecular or genetic patterns, thereby enhancing the understanding of tumor heterogeneity and improving prediction capabilities [35]. Accordingly, machine learning algorithms

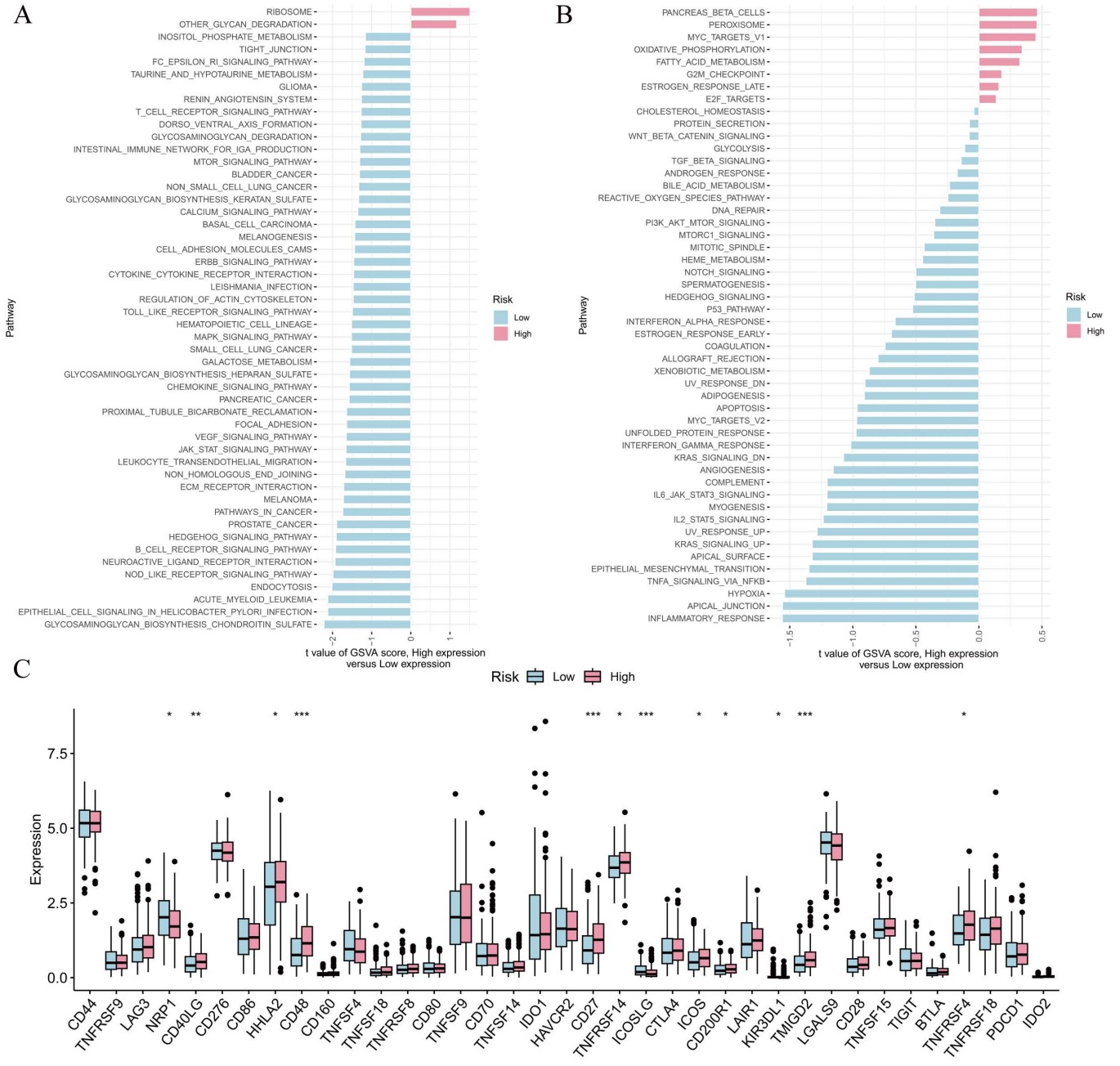

**Fig 7. Enrichment analysis of the differential genes between high- and low-risk score subgroups.** (A) KEGG enrichment analysis; (B) Hallmark enrichment analysis; (C) differential analysis of immune genes. *P < 0.05; **P < 0.01; ***P < 0.001.

were applied to construct pathomics models (RF and XGBoost models) for predicting the expression level of CLCA1 in COAD. Subsequently, the predicted accuracy of both models was evaluated. The AUC values of the ROC and PR curves demonstrated that both models exhibited adequate predictive ability. Nevertheless, the RF model demonstrated superior predictive ability when compared to the XGBoost model. This can be explained by the fact that RF is an ensemble technique that merges several decision trees, while XGBoost is a gradient-boosting method with distinct modeling approaches, resulting in variations in its prediction performance. As a result of the pathomics

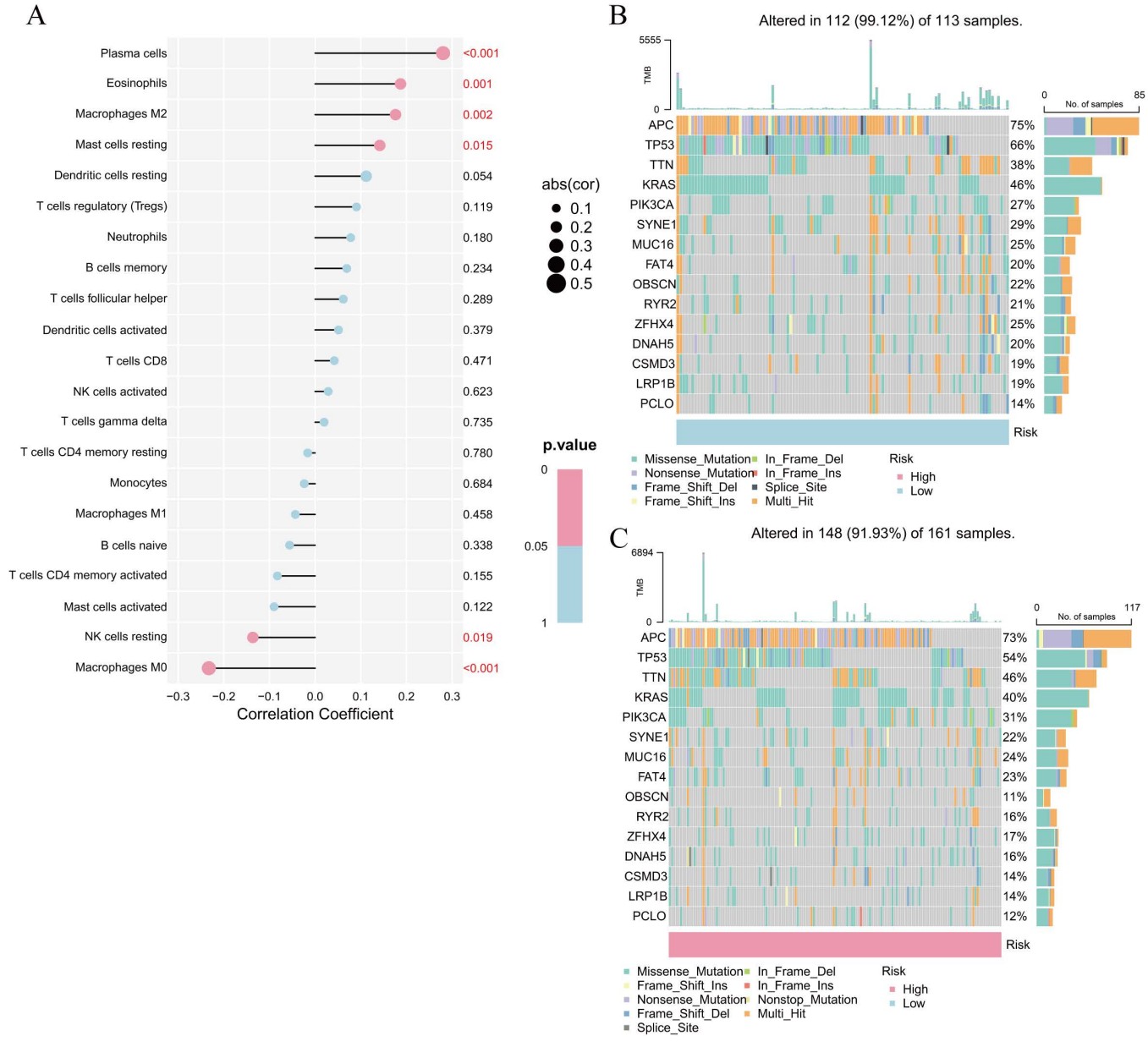

**Fig 8. Correlation analysis of immune cell abundance and gene mutation analysis.** (A) Correlation analysis of risk score with immune cell abundance; (B) Mutation analysis in the low-risk score subgroups; (C) Mutation analysis of genes in high-risk score subgroups.

models, the output risk score was associated with prognosis, suggesting that it is a reliable prognostic indicator for COAD.

Bioinformatics methods have received extensive attention from researchers for treating and determining the prognosis of patients with COAD [36]. This study thoroughly examined the association between the CLCA1 pathomics models and various factors including pathway enrichment, immune cell infiltration, gene mutations, and the differential expression of immune-related genes. Furthermore, GSVA was conducted to investigate the variations in enriched pathways between the subgroups with high- and low-risk scores. There is a close connection between enriched pathways and various malignancies and immunosuppression, which is particularly important [37–40]. Immune

-related disorders and cancers, such as melanoma, glioblastoma, head and neck cancer, lung cancer, pancreatic cancer, breast cancer, and colorectal cancer, have been associated with irregular JAK-STAT pathway activation [41,42]. Throughout the development of carcinoma, EMT contributes to cancer advancement by endowing cells with mesenchymal characteristics, which are connected to highly aggressive tumors [43]. Its involvement in the metastasis of COAD has been extensively document-ted [44].

A prior study indicates that the immune microenvironment affects the recurrence and metastasis of COAD [45]. Numerous studies emphasize the crucial role of immune cell infiltration in the progression, dissemination and immune evasion of various diseases [46]. This study found that individuals with a low-risk score had a higher number of M0 macrophages and inactive NK cells, whereas individuals with a high-risk score had increased levels of plasma cells, eosinophils, M2 macrophages, and dormant mast cells. Numerous studies have established that a significant infiltration of M0 macrophages in the tumor microenvironment may serve as an indicator of poor prognosis [47], consistent with the findings of the current study. Notably, the tumor-infiltrating immune cell environment serves as an indicator of the immune status of patients with COAD and can explain the disparity in survival outcomes between individuals with high- and low-risk scores. Moreover, the high-risk score group exhibited a marked increase in the expression levels of CD27, CD40LG, ICOS, and TMIGD2. Among these, CD27 was highly expressed in the intestinal tumor stroma, lymphoid follicles, and T-cell membranes of normal intestinal mucosa and was positively correlated with patient survival [48]. According to related research in breast cancer, the high expression of CD40LG indicates a higher survival rate [49]. These immune genes may act as potential targets for immune-related therapy of CLCA1. In agreement with our findings, the mutation analysis results of Chen et al. revealed that missense mutations are the predominant type observed in colon cancer [50]. Consistent with the findings of previous research, the current study identified that the four genes with the highest mutation frequency were APC, TP53, TTN, and KRAS [51]. Moreover, APC mutations have been reported to induce intestinal tumors by activating the Wnt signaling pathway in epithelial cells [52]. Additionally, earlier studies have demonst-rated that TP53 mutations affect its protein structure, folding, and stability, as well as its capacity for DNA binding and physiological activity, thereby facilitating tumori-genesis [53]. Similarly, mutations in the KRAS gene trigger downstream signaling pathways, such as mitogen-activated protein kinases, leading to cell growth and tumor development [54].

## Innovation and its limitations

This study revealed a strong association between the crucial gene CLCA1 and the outcomes of patients with COAD. Two pathomics models were established to predict CLCA1 expression in COAD using H&E stained pathological images, eliminating the need for additional genetic or immunohistochemical testing. The findings of this study may offer valuable insights into future personalized prediction methods and hierarchical, precise tumor treatment. However, the current study has some limitations that cannot be overlooked. To begin with, the sample size was relatively limited and may not accurately represent the racial demographic. Secondly, part of the pathological mechanism was analyzed using bioinformatics technology, which requires further validation through clinical and fundamental experiments. Future studies should concentrate on integrating large-scale population data and multi-omics approaches into clinical investigations.

## Conclusions

In conclusion, this study developed two pathomics models, the RF and XGBoost models, using the TCGA database to forecast CLCA1 expression in COAD. Additionally, the potential biological mechanisms underlying pathomics features were explored by correlating the pathomics models with transcriptomic profiles.

## Supporting information

**S1 Table. TCGA clinical data inclusion exclusions.**
(XLSX)

**S2 Table. TCGA pathological image inclusion exclusions.**
(XLSX)

**S3 Table. Grouping of high and low expression of CLCA1 gene.**
(XLSX)

**S4 Table. Clinical variables in the training and validation sets.**
(XLSX)

**S5 Table. RF model accuracy, sensitivity, specificity and Brier score values.**
(XLSX)

**S6 Table. XGBoost model accuracy, sensitivity, specificity and Brier score values.**
(XLSX)

**S7 Table. Grouping of high and low expression of risk score.**
(XLSX)

**S1 Fig. Workflow.**
(TIF)

## Author contributions

**Conceptualization:** Caiyun Yao, Yang Cao.

**Writing – original draft:** Caiyun Yao, Maotong Hu, Lingxia Zhou, Hui Chen, Yang cao.

**Writing – review & editing:** Caiyun Yao, Yang Cao.

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
