## [Decision Letter · Decision Letter 0]

PONE-D-25-02453Establishment of two pathomic-based machine learning models to predict CLCA1 expression in colon adenocarcinomaPLOS ONE

Dear Dr. cao,

Thank you for submitting your manuscript to PLOS ONE. After careful consideration, we feel that it has merit but does not fully meet PLOS ONE’s publication criteria as it currently stands. Therefore, we invite you to submit a revised version of the manuscript that addresses the points raised during the review process.

We look forward to receiving your revised manuscript.

Kind regards,

Jinhui Liu

Academic Editor

PLOS ONE

**Journal Requirements:**

1. When submitting your revision, we need you to address these additional requirements. Please ensure that your manuscript meets PLOS ONE's style requirements, including those for file naming. The PLOS ONE style templates can be found at https://journals.plos.org/plosone/s/file?id=wjVg/PLOSOne_formatting_sample_main_body.pdf and https://journals.plos.org/plosone/s/file?id=ba62/PLOSOne_formatting_sample_title_authors_affiliations.pdf 2. Please note that PLOS ONE has specific guidelines on code sharing for submissions in which author-generated code underpins the findings in the manuscript. In these cases, we expect all author-generated code to be made available without restrictions upon publication of the work. Please review our guidelines at https://journals.plos.org/plosone/s/materials-and-software-sharing#loc-sharing-code and ensure that your code is shared in a way that follows best practice and facilitates reproducibility and reuse. 3. PLOS ONE now requires that authors provide the original uncropped and unadjusted images underlying all blot or gel results reported in a submission’s figures or Supporting Information files. This policy and the journal’s other requirements for blot/gel reporting and figure preparation are described in detail at https://journals.plos.org/plosone/s/figures#loc-blot-and-gel-reporting-requirements and https://journals.plos.org/plosone/s/figures#loc-preparing-figures-from-image-files. When you submit your revised manuscript, please ensure that your figures adhere fully to these guidelines and provide the original underlying images for all blot or gel data reported in your submission. See the following link for instructions on providing the original image data: https://journals.plos.org/plosone/s/figures#loc-original-images-for-blots-and-gels.   In your cover letter, please note whether your blot/gel image data are in Supporting Information or posted at a public data repository, provide the repository URL if relevant, and provide specific details as to which raw blot/gel images, if any, are not available. Email us at plosone@plos.org if you have any questions.

**Additional Editor Comments:**

Authors should revise according to the suggestions of reviewers. The modifications should be marked. A point to point response letter is needed.

Reviewers' comments:

Reviewer's Responses to Questions

**Comments to the Author**

1. Is the manuscript technically sound, and do the data support the conclusions?

Reviewer #1: Partly

2. Has the statistical analysis been performed appropriately and rigorously? 

Reviewer #1: Yes

3. Have the authors made all data underlying the findings in their manuscript fully available?

Reviewer #1: Yes

4. Is the manuscript presented in an intelligible fashion and written in standard English?

Reviewer #1: No

5. Review Comments to the Author

**Reviewer #1:**  the study explores the prognostic value of CLCA1 in COAD and its potential biological mechanisms by combining pathomics and transcriptomics data, offering new insights for precision medicine in COAD. However, some limitations impair the overall quality. I suggest the author make appropriate modifications.

1. It is recommended to include a flowchart of the study to visually present the research design, data processing, and analysis steps, enhancing the readability and logical clarity of the study.

2. The authors state " CLCA1 expression was lower in cancer tissues of various tumors, including COAD, according to the differential analysis results (P<0.001)". However, p < 0.001 implies that the authors have used only nominal p-values without multiple hypothesis correction. The authors should use FDR or similar to control false-positive rate.

3. Please provide the full definition of abbreviations the first time they are used, and include the abbreviation in parentheses.

4. Some previous studies have utilized similar bioinformatics and statistical methods (PMID: 39702250; PMID: 37437243; PMID: 36213111). While referencing these studies may reduce the novelty of the work to some extent, I believe it would greatly benefit the readers by providing a more comprehensive understanding of the context of the current study.

5. I recommend that the manuscript undergo professional language editing and proofreading to improve clarity.

6. PLOS authors have the option to publish the peer review history of their article (what does this mean? ). If published, this will include your full peer review and any attached files.

**Do you want your identity to be public for this peer review?** For information about this choice, including consent withdrawal, please see our Privacy Policy .

Reviewer #1: No

---

## [Author Response · Author response to Decision Letter 1]

24 Jun 2025

Dear Editor,

We would like to express our sincere gratitude to you and the reviewers for the insightful comments and valuable suggestions provided during the review of our manuscript titled " Establishment of two pathomic-based machine learning models to predict CLCA1 expression in colon adenocarcinoma."We have carefully considered all the feedback and have made revisions to improve the quality of our manuscript. Below, we provide a detailed response to reviewers' comments and outline the specific changes made to the manuscript.

Response to Reviewer 1:

Comment 1: It is recommended to include a flowchart of the study to visually present the research design, data processing, and analysis steps, enhancing the readability and logical clarity of the study.

Response 1: Thank you for your suggestion. We developed a flowchart encompassing pathomics establishment and prognostic analysis to clarify the research process.

Comment 2: The authors state " CLCA1 expression was lower in cancer tissues of various tumors, including COAD, according to the differential analysis results (p < 0.001)". However, p < 0.001 implies that the authors have used only nominal p-values without multiple hypothesis correction. The authors should use FDR or similar to control false-positive rate.

Response 2: Thank you for pointing this out. We applied the Benjamini-Hochberg adjustment to our P values to enhance the reliability of our statistical results.

Comment 3: Please provide the full definition of abbreviations the first time they are used, and include the abbreviation in parentheses.

Response 3: We appreciate your reminder. We reviewed the entire document and supplemented any incomplete abbreviations.

Comment 4: Some previous studies have utilized similar bioinformatics and statistical methods (PMID: 39702250; PMID: 37437243; PMID: 36213111). While referencing these studies may reduce the novelty of the work to some extent, I believe it would greatly benefit the readers by providing a more comprehensive understanding of the context of the current study.

Response 4: Thank you for your suggestions! The references you suggested are beneficial to us. We have read through all the references and learned a great deal. They played a key role in improving the quality of our article. Therefore, we have cited them in the revised manuscript. Thanks once again.

Comment 5: I recommend that the manuscript undergo professional language editing and proofreading to improve clarity.

Response 5: We appreciate your reminder. We have carefully proofread the entire manuscript and corrected all typographic and grammatical errors.

Please review these changes.

We hope these modifications meet the reviewer's requirements, and we thank you again for your valuable feedback.

Best regards,

Yours sincerely,

Yang Cao

Yiwu central hospital, Jinhua, Zhejiang, China

Pathology Department. Yiwu, 322000, China

Email: 1814218039@qq.com

---

## [Decision Letter · Decision Letter 1]

Establishment of two pathomic-based machine learning models to predict CLCA1 expression in colon adenocarcinoma

PONE-D-25-02453R1

Dear Dr. cao,

We’re pleased to inform you that your manuscript has been judged scientifically suitable for publication and will be formally accepted for publication once it meets all outstanding technical requirements.

Kind regards,

Jinhui Liu

Academic Editor

PLOS ONE

Additional Editor Comments (optional):

I think this manuscript was well organized and it could be accepted.

Reviewers' comments:

Reviewer's Responses to Questions

**Comments to the Author**

1. If the authors have adequately addressed your comments raised in a previous round of review and you feel that this manuscript is now acceptable for publication, you may indicate that here to bypass the “Comments to the Author” section, enter your conflict of interest statement in the “Confidential to Editor” section, and submit your "Accept" recommendation.

Reviewer #1: All comments have been addressed

2. Is the manuscript technically sound, and do the data support the conclusions?

Reviewer #1: Yes

3. Has the statistical analysis been performed appropriately and rigorously? 

Reviewer #1: Yes

4. Have the authors made all data underlying the findings in their manuscript fully available?

Reviewer #1: Yes

5. Is the manuscript presented in an intelligible fashion and written in standard English?

Reviewer #1: Yes

6. Review Comments to the Author

Reviewer #1: The author's modifications have effectively resolved the previous issues, and I have no other problems with the current version

7. PLOS authors have the option to publish the peer review history of their article (what does this mean? ). If published, this will include your full peer review and any attached files.

**Do you want your identity to be public for this peer review?** For information about this choice, including consent withdrawal, please see our Privacy Policy .

Reviewer #1: No

---

## [Editor Report · Acceptance letter]

PONE-D-25-02453R1

PLOS ONE

Dear Dr. cao,

I'm pleased to inform you that your manuscript has been deemed suitable for publication in PLOS ONE. Congratulations! Your manuscript is now being handed over to our production team.

Kind regards,

on behalf of

Dr. Jinhui Liu

Academic Editor

PLOS ONE